# Semantic Terrain Classification for Off-Road Autonomous Driving

**Amirreza Shaban**,* **Xiangyun Meng**\*, **JoonHo Lee**\*, **Byron Boots, Dieter Fox**

University of Washington

**Abstract:** Producing dense and accurate traversability maps is crucial for autonomous off-road navigation. In this paper, we focus on the problem of classifying terrains into 4 cost classes (free, low-cost, medium-cost, obstacle) for traversability assessment. This requires a robot to reason about both semantics (what objects are present?) and geometric properties (where are the objects located?) of the environment. To achieve this goal, we develop a novel *Bird's Eye View Network* (BEVNet), a deep neural network that directly predicts a local map encoding terrain classes from sparse LiDAR inputs. BEVNet processes both geometric and semantic information in a temporally consistent fashion. More importantly, it uses learned prior and history to *predict* terrain classes in unseen space and into the future, allowing a robot to better appraise its situation. We quantitatively evaluate BEVNet on both on-road and off-road scenarios and show that it outperforms a variety of strong baselines.
Website: https://sites.google.com/view/terrain-traversability/home.

**Keywords:** Off-road Driving, Autonomous Driving, Deep Learning, Perception

## 1 Introduction

While there has been great recent interest in the development of autonomous vehicles, the vast majority of this work has focused on *on-road* and *urban* driving. However, a wide range of application areas including defense, agriculture, conservation, and search and rescue could benefit from autonomous *off-road* vehicles that can operate in complex, natural terrain. In such environments, understanding the *traversability* of the terrain surrounding the vehicle is crucial for successful planning and control. Per-

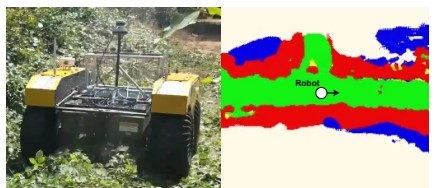

Figure 1: **Example target scenario.** BEVNet can be run on the Clearpath Warthog to generate semantic map of offroad envrionments.

ceiving whether the terrain is traversable from sparse LiDAR data can be a challenging problem as off-road terrain is often characterized by rapid changes to the ground plane, heavy vegetation, overhanging branches, and negative obstacles. In other words, a successful off-road robot must reason about both the geometric and semantic content of its surroundings to determine what terrain is traversable and what is non-traversable.

In this work, we formulate traversability estimation as a semantic terrain classification problem [1, 2]. The motivation is to unify the semantics (what objects are present?) and geometry (where are the objects located?) of the terrain into a single cost ontology. From the semantic perspective, objects such as large rocks and tree trunks are non-traversable, whereas gravel, grasses, and bushes are all traversable by an off-road vehicle [3] but with increasing difficulty. From the geometric perspective, overhanging obstacles can be ignored, and objects of the same semantic class may vary in their traversability depending on their heights (e.g., tall bushes vs. short bushes). To this end, we use a discrete set of traversability levels for the ease of grouping the semantic classes by their traversability while allowing us to adjust the traversability levels of specific instances based on their geometry.

---

*Equal Contribution. Contact at {`ashaban, xiangyun, joonl4`}@cs.washington.edu

5th Conference on Robot Learning (CoRL 2021), London, UK.

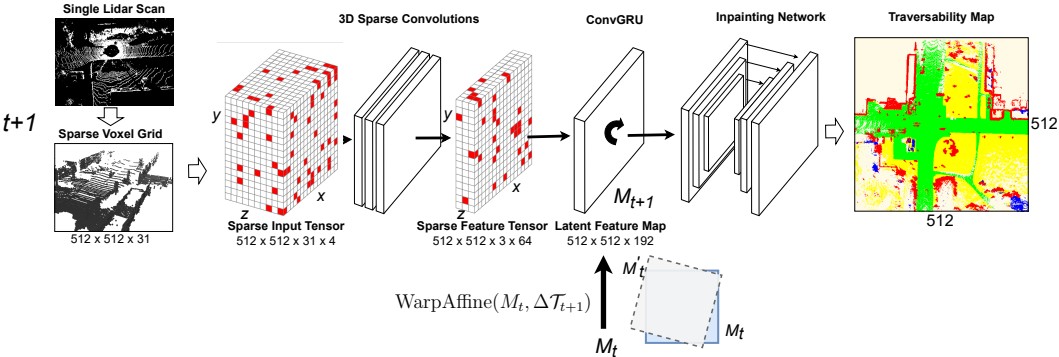

Figure 2: The network architecture of BEVNet. The incoming LiDAR scan is first discretized into a sparse voxel grid, which is then fed into a sequence of sparse convolution layers to compress the $z$ dimension. The compressed sparse feature tensor is aggregated over time via the ConvGRU unit. We use differentiable affine warping to align the latent feature map with the current odometry frame. Finally, the inpainting network "inpaints" the latent map to output a dense traversability map.

An effective terrain classification system should efficiently 1) aggregate observations over time [1, 4] with noisy odometry [5, 6], 2) reason about the partially seen or even yet to be seen parts of the environment [7, 8, 9], and 3) detect overhanging structures such as tree branches, tunnels, and power-lines [10, 11]. While previous work has addressed each of these issues individually, the aforementioned challenges are related, and solving each one should benefit the others.

In this paper, we propose a *Bird's Eye View Network* (BEVNet), a recurrent neural network that directly predicts terrain classes in the form of a 2D grid around the robot from LiDAR scans. As shown in Figure 2, our model has three main parts: 1) a 3D sparse convolution sub-network to process the voxelized point cloud, 2) a Convolutional Gated Recurrent Unit (ConvGRU) which uses convolutional layers in a gated recurrent unit [12] to aggregate the 3D information, 3) an efficient 2D convolutional encoder-decoder based on [13] that simultaneously inpaints the empty spaces and projects the 3D data into the 2D Bird's Eye View (BEV) map. To train the model, we use both past and future labeled LiDAR scans to build a complete 3D semantic point cloud and build the ground-truth 2D traversability map. Previous work [10] utilizes a collection of collapsible cube structures with ground/overhang classification to remove irrelevant overhangs based on their gaps, but such rule-based filtering lacks generalization when it is difficult to estimate accurate ground levels from sparse LiDAR scans. In comparison, our model is trained with a *ground-truth BEV map* built from fully observed and labeled environments, which allows accurate ground-level estimation for learning. The network then *learns* to detect and remove overhanging obstacles from sparse LiDAR scans without an explicit filtering mechanism.

We make several contributions and empirical observations. We proposed a novel framework to build the BEV costmap by simultaneously 1) aggregating observations over time, 2) predicting the unseen areas of the map, and 3) filtering out irrelevant obstacles like overhanging tree branches that do not affect traversability. Experimental results on SemanticKITTI [14] and RELLIS-3D [15] show that BEVNet outperforms strong baselines in both on-road and off-road settings.

## 2 Related Work

Since most prior literature on perception for autonomous driving [16, 17, 18, 19, 20, 21, 22, 23] focus on *urban* environments and structure inherent in cities and road networks, we compare our system to works that share the mutual components to ours, namely traversability analysis [24, 25, 26, 27], semantic mapping [1, 17, 18, 19, 23, 21], recurrency handling [1, 17, 19], and semantic scene completion [7, 8, 9].

**Traversability Analysis and Semantic Mapping.** Traversability analysis and semantic mapping are crucial for off-road autonomy [25, 28, 29]. Traversability may be analyzed based on various criteria, including surface roughness [24], negative obstacles [25], and terrain classification [26, 27]. The traversability information is usually projected into a Bird's Eye View (BEV) map, which stores local traversability and semantic information in a topdown 2D grid [30]. While prior works such as [16, 17, 18] utilize a high definition map assumed *a priori*, such a map is expensive to produce,

and therefore our system instead constructs one online. Similarly, Casas et al. [19], Wu et al. [23] produces online semantic maps for urban autonomous driving but prioritize mapping of lanes and road objects. In comparison, our system focuses on a broader categorization of semantics based on terrain traversability. Additional works generate semantic maps from images, where Philion and Fidler [21] produce BEV maps from RGB images end-to-end, while Maturana et al. [1] apply a bayesian filter. Our system instead learns to produce semantic maps from LiDAR scans. We also compare our approach to LiDAR segmentation where the segmented point cloud from networks such as [31, 32] can be projected onto a BEV map. We evaluate this in detail in Sec. 5.

**Temporal Aggregation** Temporally consistent semantics is crucial for stably mapping the environment. Recent work [17, 19] directly concatenates the past 10 scans as its input for memory efficiency. In comparison to our system, the author's recurrent neural network (RNN) is specifically designed for semantic occupancy forecasting, whereas our recurrent architecture accumulates features sequentially to better estimate the traversability of the current surrounding terrain. Maturana et al. [1] accumulate information via Bayes filtering, which in our system is also replaced by the RNN. There are various available architectures for handling recurrency[4, 33, 34, 35, 36, 37, 38], of which our system utilizes ConvGRU to accumulate 2D BEV maps.

**Semantic Scene Completion (SSC)** The goal of SSC is to generate a complete 3D scene given a single LiDAR scan. [7, 8, 9] utilize information from semantic segmentation to complete the scene, whereas our system learns to directly predict the completed scene and therefore does not require segmentation from a secondary network or ground truth labels. In addition, our system performs point cloud projection and scene completion in 2D simultaneously, as the main task in our work is to produce a 2D BEV map. We evaluate our system's ability to complete scenes by comparing our model against [7], the details can be found in Sec. 5.

## 3    Method

### 3.1    Overview

We consider a mobile robot with a $360°$ LiDAR mounted at its top. In order for the robot to navigate efficiently and safely in a new environment (either on-road or off-road), the robot builds an *online traversability map* around itself. The traversability map resembles a conventional occupancy map as well as the semantic map from [1], where each cell stores a probability distribution of traversability labels. In this work, we use four levels of traversability: *free*, *low-cost*, *medium-cost*, and *lethal*. The number of traversability levels can be trivially extended if so desired. The traversability map is inside the robot's odometry frame, so that the robot is always at the center, with its heading pointing to the east. The traversability map is converted to a costmap by mapping each traversability level to the corresponding cost value via a lookup table. The converted costmap can be easily interfaced with a local planner [39] or a global planner (e.g., A*) for finding the least-cost path to a goal.

We adopt a supervised-learning approach to predict this traversability map. We start with building a traversability dataset from LiDAR segmentation datasets [14, 15] via a traversability-aware projection procedure. Then, we introduce BEVNet, a recurrent neural network that takes the current LiDAR scan and utilizes its history to build a dense traversability map. In the following sections, we will describe each component in detail.

### 3.2    Building a Traversability Dataset

Recent work [17, 1] focuses on on-road driving where reasoning about a large number of fine-grained semantic classes is necessary. Here we consider a more general driving paradigm where we simply care about the *traversability* of the surrounding terrain. This makes our model applicable to both on-road and off-road driving. Given a dataset with semantically labeled LiDAR scans, we convert it to a traversability dataset via the following procedure (illustrated in Figure 3).

**Scan Aggregation.** For each scan, we aggregate it with the past $t$ and the future $t$ scans with stride $s$ to construct a larger point set. We set $t$ to a large enough number (e.g., 71) to obtain dense traversability information for a large area around the robot. These parameters may be tuned depending on the vehicle speed and density of the LiDAR points.

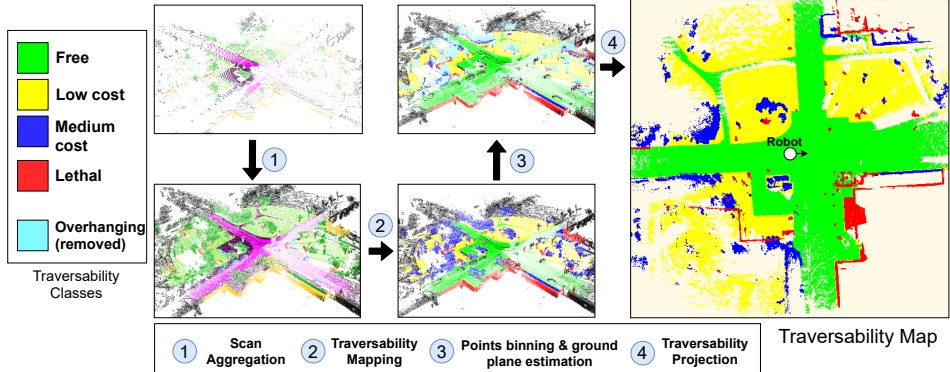

Figure 3: Process for generating the traversability dataset on SemanticKITTI. Single scan labels are aggregated to form complete scenes (scan aggregation), and their semantics are remapped to our 4 class ontology (traversability mapping), legend defined in the figure. The remapped scans are then filtered with ground estimation to remove overhanging points (points binning and ground plane estimation). Finally, the filtered points are projected to a traversability map (traversability projection).

**Traversability Mapping.** We map the semantic classes into our 4-level ontology. The general principle is to map semantic classes with similar costs to the same traversability label. For example, *cars* and *buildings* are mapped to *lethal*, whereas *mud* and *grass* are mapped to *low-cost*. Detailed mapping can be found in the supplementary.

**Points Binning and Ground Height Estimation.** For each point in the aggregated scan, we do a down projection to find its location $x, y$ on the traversability map. Hence each $x, y$ location of the map contains a pillar of points. We estimate the ground height map by running a mean filter kernel over the lowest $z$ coordinates of the points labeled as *free* and *low-cost* at each $x, y$ location in the map. This height map is used as a reference for final traversability projection.

**Traversability Projection.** For each pillar of points, we filter out overhanging obstacles by removing points that are above the local ground level by a certain threshold because they will not collide with the robot. Additionally, we adjust the traversability level of certain points based on their height above the ground level and the mobile capabilities of the robot. i.e. points labeled as *medium-cost* but very close to the local ground level can be deemed negligible for a large off-road vehicle and hence remapped to *low-cost* as other nearby points. Finally, we take the class of the least traversable point (i.e., most difficult) at each $x, y$ location as the final traversability label.

### 3.3 Feature Extraction via Sparse Convolution with Z Compression

The architecture of BEVNet is shown in Figure 2. An input LiDAR scan is first discretized into a $512 \times 512 \times 31$ grid with a resolution of $0.2m$. We perform sparse discretization so that only occupied voxels are preserved. Each voxel contains a 4-dimensional feature $f = \frac{1}{n} \sum_{i=1}^{n} [x_i, y_i, z_i, r_i]$, which is the average of the coordinates and remission values of the points inside the voxel. This sparse voxel grid is fed into a sequence of sparse convolution layers, which compress the $z$ dimension via strided convolutions. We keep $x$ and $y$ dimensions unchanged. The output of the sparse convolution layers is a sparse feature tensor $S$ of size $512 \times 512 \times C$, where $C$ is the feature dimension.

### 3.4 Temporal Aggregation of Sparse Feature Maps

A single LiDAR scan becomes increasingly sparse as the distance increases, making it difficult to classify the traversability level for areas far away from the robot. Contrary to classical SLAM that aggregates LiDAR measurements over time via a hand-engineered Bayesian update rule [40], we let the network learn to aggregate the sparse feature maps from past LiDAR scans via a Convolutional Gated Recurrent Unit (ConvGRU). The ConvGRU maintains a 2D latent feature map $M$ that shares the same coordinate system and dimensions as the final traversability map. The latent feature map $M$ is updated as

$$M_{t+1} = \text{ConvGRU}(\text{WarpAffine}(M_t, \Delta \mathcal{T}_{t+1}), S_{t+1}),$$

where $\Delta \mathcal{T}_{t+1}$ is the relative transform of the robot's odometry frame from $t$ to $t+1$. The WarpAffine operation transforms the latent feature map $M_t$ from the previous odometry frame to the cur-

rent odometry frame so that the features from $M_t$ and $S_{t+1}$ are spatially aligned. Note that the WarpAffine operation is differentiable to allow the gradients to backpropagate through time.

### 3.5 Traversability Inpainting

Since the ConvGRU only aggregates sparse feature tensors, $M$ contains little information for areas where there is no LiDAR point. Instead of treating the no-hit area as unknown, we let the network fill in the empty space by leveraging the local and global contextual cues via the *Inpainting Network*. The inpainting network is a fully convolutional network inspired by FCHardNet [13] which is originally designed for fast image segmentation. It consists of a sequence of downsampling and upsampling layers with skip connections, making it effective for capturing local and global contextual information for predicting what is missing.

## 4 Implementation Details

We build the traversability datasets from SemanticKITTI [14] and RELLIS-3D [15] to evaluate BEVNet in both on-road and off-road scenarios. For SemanticKITTI we aggregate 71 frames with stride 2 to generate a single traversability map. For RELLIS-3D we aggregate 141 frames with stride 5. Both datasets provide per-frame odometry, which we use for the differential warping layer in the ConvGRU. The traversability maps have a size of $102.4m \times 102.4m$ with a resolution of $0.2m$. Note that the traversability maps contain additional "unknown" class marking regions that have never been observed. For additional details on network training and data augmentation please refer to section 4 in the supplementary material.

## 5 Experiments

We conduct both quantitative and qualitative studies on SemanticKITTI (on-road) and RELLIS-3D (off-road) datasets. We trained a separate model for each dataset. We compare with a variety of baselines, ranging from LiDAR segmentation to scene completion on the validation sequences. We also perform an ablation study to better understand the contribution of recurrence, and how our model behaves on the two datasets that have very different characteristics.

### 5.1 Evaluation Metrics

We use the mean Intersection of Union (mIoU) [13], a widely used metric for image segmentation, as the quantitative measure of the prediction accuracy. Note that our model predicts an additional "unknown" class to improve the visual consistency, and we exclude the "unknown" class in the evaluation. To better understand our model's capability of predicting the future, we report mIoUs in three modes: **seen**, **unseen**, and **all**. In the "seen" mode, we do not include ground truth labels obtained from future frames, effectively excluding any future predictions. For the "unseen" model we only include the future predictions. In the "all" scenario, we evaluate both.

### 5.2 Comparison with LiDAR Segmentation with Temporal Aggregation

A strong baseline for building a traversability map is to perform semantic segmentation of the incoming LiDAR scan, project it down to obtain a 2D sparse traversability map, and aggregate the traversability maps over time. To compare with this approach, we choose Cylinder3D [31] (finetuned on our 4-class ontology) as the LiDAR segmentation network for its strong performance and use the same projection procedure in Sec 3 on the input LiDAR scan to obtain the single-frame traversability map. We perform the temporal aggregation by tracking the categorical distribution of traversability via a uniform Dirichlet prior. To do so, we keep a counter map $M_C$ of size $H \times W \times 4$ (initialized to zeros). It is of the same size as the traversability map except that the last dimension counts the traversability labels observed so far. We update $M_C$ incrementally. For each incoming single-frame traversability map, we warp $M_C$ to the current odometry frame via bilinear interpolation, and increment the counts by adding the one-hot version of the incoming single-frame map. The actual traversability label can be obtained by taking the *argmax* of the last dimension of $M_C$.

|  | SemanticKITTI | | | RELLIS-3D | | |
|---|---|---|---|---|---|---|
|  | All | Seen | Unseen | All | Seen | Unseen |
| BEVNet-S | 0.416 | 0.465 | 0.308 | 0.559 | 0.518 | 0.545 |
| **Clean Odometry** | | | | | | |
| BEVNet-TA | 0.468 | 0.534 | 0.335 | 0.615 | 0.605 | 0.586 |
| BEVNet-R | **0.535** | 0.625 | **0.354** | **0.644** | 0.621 | **0.621** |
| Cylinder3D-TA | 0.465 | 0.655 | N/A | 0.411 | 0.568 | N/A |
| Cylinder3D-TA-3D w/o ray tracing | 0.482 | **0.660** | N/A | 0.408 | **0.649** | N/A |
| Cylinder3D-TA-3D w/ ray tracing | 0.471 | 0.646 | N/A | 0.384 | 0.609 | N/A |
| **Noisy Odometry** | | | | | | |
| BEVNet-TA | 0.379 | 0.415 | 0.310 | 0.452 | 0.372 | 0.560 |
| BEVNet-R | **0.468** | **0.529** | **0.343** | **0.614** | **0.572** | **0.616** |
| Cylinder3D-TA | 0.342 | 0.455 | N/A | 0.318 | 0.435 | N/A |
| Cylinder3D-TA-3D w/o ray tracing | 0.373 | 0.479 | N/A | 0.347 | 0.517 | N/A |
| Cylinder3D-TA-3D w/ ray tracing | 0.369 | 0.478 | N/A | 0.331 | 0.495 | N/A |

Table 1: Mean IoU of different methods on SemanticKITTI and RELLIS-3D.

**Results on SemanticKITTI.** In the left half of Table 1 we compare the performance of BEVNet-Recurrent (BEVNet-R) with Cylinder3D+Temporal Aggregation (C3D-TA) on the SemanticKITTI validation set. When only considering what has been observed so far ("seen") and clean odometry, C3D-TA is better than BEVNet-R. This shows that LiDAR segmentation with accurate temporal aggregation can work very well in structured environments such as urban driving. When evaluated on the full groundtruth ("full"), BEVNet-R outperforms C3D-TA because C3D-TA cannot predict the future traversability. When evaluating on noisy odometry, BEVNet-R surpasses C3D-TA for both "seen" and "full" test scenarios. BEVNet-R uses learned recurrency to "fix" small errors in odometry and to adaptively forget history in case the error is too large. In comparison, C3D-TA solely uses the provided odometry to aggregate information, which may result in large misalignment as errors compound over time.

**Results on RELLIS-3D.** The results on RELLIS-3D (right half of Table 1) share a similar trend as those in SemanticKITTI, except that BEVNet-R consistently outperforms C3D-TA with a larger gap. This suggests that off-road environment is more challenging, where accurate LiDAR segmentation is hard to obtain due to the lack of environmental structure. Indeed, Cylinder3D only achieves a 64.1 mIoU on RELLIS-3D for LiDAR segmentation, which is lower than the 87.9 mIoU on SemanticKITTI. Interestingly, noisy odometry has a smaller impact on BEVNet-R. We hypothesize that it is because the RELLIS-3D dataset contains less clutter and occlusion so BEVNet-R does not rely heavily on the history for traversability prediction.

**Comparison with 3D Semantic Temporal Aggregation.** We additionally provide a baseline that aggregates the points in 3D before projection (Cylinder3D-TA-3D) using Octomap [41]. The results are included in Table 1. C3D-TA-3D in general works better than C3D-TA and can even outperform BEVNet-R in the seen area. However, since it is not able to predict the future, its mIoU on the full scene is poor. It is significantly slower (less than 1 fps), and is susceptible to odometry noise like C3D-TA. We optionally enable ray tracing in Octomap to remove dynamic obstacles. While quantitatively this does not make a large difference, there is a clear distinction in the qualitative comparison described next.

### 5.2.1 Qualitative Results

In left half of Figure 4, we highlight that BEVNet-R can preserve small dynamic objects such as bicyclists better than C3D-TA. Hand-engineered temporal aggregation is prone to treating small dynamics objects as noise and ignoring them. In comparison, BEVNet can learn to keep small dynamic objects, while preserving smoothness in static regions. The right half shows the impact of noisy odometry. We can see large misalignment and smear artefacts for C3D-TA, whereas BEVNet-R produces significantly cleaner output. In Figure 5 we provide a comparison between BEVNet-R, C3D-TA, and C3D-TA-3D. C3D-TA-3D can preserve more details in the map, but without ray tracing, it fails to clear the residues left by dynamic obstacles such as cars and humans. While turning on ray tracing [41] improves the results, it also erroneously clears some of the ground points due to shallow LiDAR incident angles. In comparison, BEVNet-R does not suffer from these shortcomings and predicts a more complete area. Finally, in Figure 6 we visualize examples on both SemanticKITTI and RELLIS-3D. In general, BEVNet-R shows strong performance in predicting future traversability. It learns to predict whole cars, alley entrances, and trail paths with extremely sparse LiDAR points.

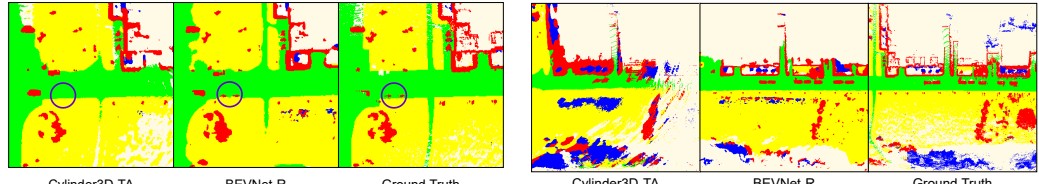

Figure 4: Qualitative comparison of Cylinder3D-TA and BEVNet-R. **Left:** BEVNet is better at preserving small fast-moving objects such as bicyclists (highlighted by the blue circles), which the hand-engineered update rule tends to ignore. (Maps are 50% zoomed in). **Right:** When noise is injected into the odometry, BEVNet-R still predicts a clean map albeit with fewer details, while Cylinder3D+TA fails to do this, resulting in a blurry, inaccurate map.

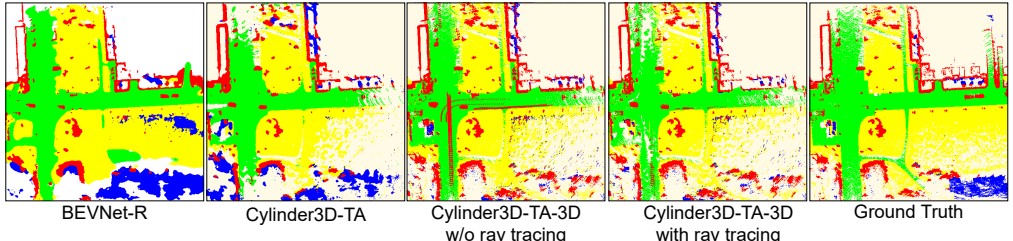

Figure 5: Comparing BEVNet-R, Cylinder3D-TA and Cylinder3D-TA-3D. While Cylinder3D-TA-3D produces more detailed maps, it introduces additional artefacts due to dynamic obstacles. See text for more details.

### 5.2.2 Ablation Study

We conduct our ablation study on three variants of BEVNet: BEVNet-Single (BEVNet-S), BEVNet-Single+Temporal Aggregation (BEVNet-TA), and BEVNet-Recurrent (BEVNet-R). We aim to answer three questions: 1) is learned recurrence better than temporal aggregation? 2) does history help predict the future? and 3) where should the information be aggregated in the network? We answer these questions through a set of experiments on both SemanticKITTI and RELLIS-3D datasets.

**Is learned recurrence better than temporal aggregation?** When evaluated on the full ground truth, we observe that BEVNet-R consistently outperforms BEVNet and BEVNet-TA on both on-road and off-road scenarios (Table 1). Notably, BEVNet-TA also outperforms BEVNet, which shows that any form of recurrence is beneficial. In particular, we observe that the learned recurrence makes the best use of the temporal information in comparison to the hand-engineered TA. When noisy odometry is introduced we observe the same trend as discussed in Sec. 5.2, where BEVNet-R shows robustness to noise and outperforms BEVNet-TA.

**Does history help predict the future?** In Table 1, we can see that any form of recurrence that accumulates history helps with predicting the unseen area. In particular, BEVNet-R outperforms both BEVNet-S and BEVNet-TA, showing that the learned recurrence can better predict the unseen area than a single-frame model or a hand-engineered temporal aggregation approach.

**Where to put ConvGRU?** Recurrence may be applied right after the sparse convolutions (*early aggregation*) or may be applied after the 2D inpainting network (*late aggregation*). We compare the two options on the SemanticKITTI dataset with clean odometry in Table 2. Note that our model is trained with clean odometry. Early aggregation yields better results than late aggregation. This

|  | mIoU |
|---|---|
| Early Aggregation | **0.535** |
| Late Aggregation | 0.479 |

Table 2: Effect of GRU location in the network.

is because when early aggregation is applied the inpainting network has access to temporally fused information and therefore is given more information to complete the scene and maintain temporal consistency across scans. Furthermore, we may infer that if late aggregation is applied, it is more difficult for the recurrent network to learn to correct the odometry as it is given completed scenes with potentially noisy information instead of the sparse feature maps.

### 5.3 Comparison and Discussion with other Related Work

Several recent works have focused on semantic understanding of scenes from sparse LiDAR scans [7, 9]. They solve related, but slightly different problems. They produce smaller maps, and do not

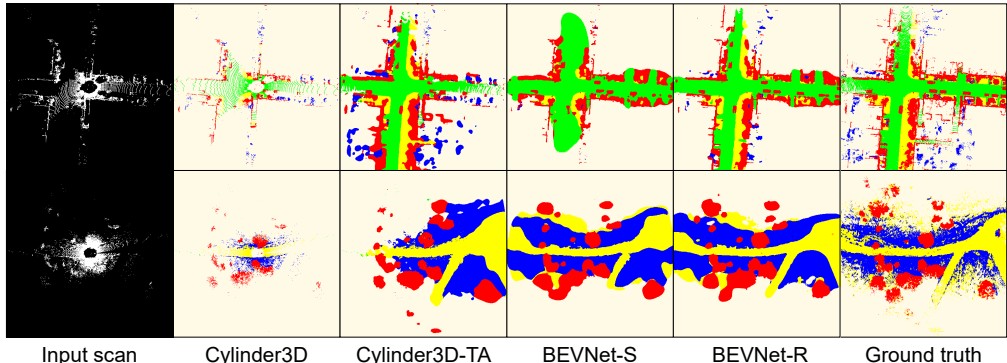

| Input scan | Cylinder3D | Cylinder3D-TA | BEVNet-S | BEVNet-R | Ground truth |

Figure 6: Qualitative results of our method on SemanticKITTI [14] (top) and RELLIS-3D [15] (bottom). Learned recurrency in our end-to-end network can preserve previously observed information while predicting future observations for the occluded areas.

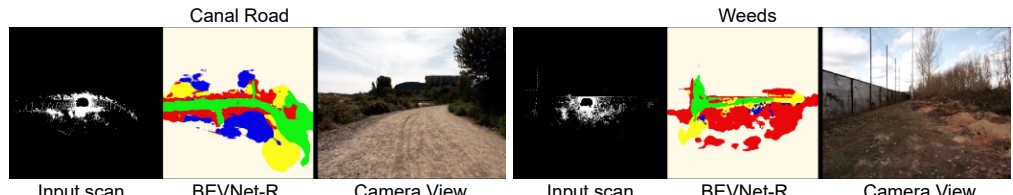

| Input scan | BEVNet-R | Camera View | Input scan | BEVNet-R | Camera View |

Figure 7: Qualitative results of BEVNet on data collected on a Clearpath Warthog. Each item shows LiDAR scan input, BEVNet-R's output, and frontal camera view (for reference), respectively.

perform temporal aggregation. Due to space limits we refer the reader to Sec.4 of the supplementary for a detailed discussion.

## 5.4 Real Robot Experiments

In Figure 7, we show that BEVNet-R trained on SemanticKITTI and RELLIS-3D can generalize to novel environments on a ClearPath Warthog robot. The robot is equipped with an OS1-64 LiDAR which is fed into BEVNet-R to classify the terrains. The first environment *Canal Road*, is a dirt trail with light vegetation, whereas the second environment *Weeds*, is scattered with grass and tree branches and is uneven. In both scenarios, BEVNet-R is able to predict a complete traversability map with sparse lidar inputs, and reason about the traversability of surroundings using the semantic and geometric features. More details can be found on the website.

## 6 Conclusion

We propose BEVNet, a framework that classifies terrain traversability in a local region around a mobile robot with the aim of helping the robot navigate in a novel on-road or off-road environment. BEVNet addresses a number of challenges in a unified architecture, namely: 1) it learns to aggregate sparse LiDAR information over time, 2) it learns to reason about traversability that involves both geometric and semantic understanding of the environment, and 3) it learns to fill in the unknown space where there are no LiDAR hits, and thus provides the robot with a more complete understanding of its surroundings. Most notably, BEVNet can leverage past information to *better predict the future*. We believe BEVNet provides an important step towards robot autonomy on complex terrains where a prior map is unavailable.

**Limitations and Future Work.** Since BEVNet is data-driven, it may overfit to certain types of environments if the training data is not diverse enough. Moreover, due to the fact that the publicly available datasets do not contain rough terrains, we do not take the roughness of terrains into account at the moment. However, BEVNet can be extended to predict additional quantities such as an elevation map to be fused with the current semantic costmap for more accurate traversability estimation. In the future, we would like to extend the capability of BEVNet and test it in more challenging off-road scenarios.

**Acknowledgments**

This work was supported in part by ARL SARA CRA W911NF-20-2-0095.

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
