# OpenReview forum: "Semantic Terrain Classification for Off-Road Autonomous Driving"
_robot-learning.org/CoRL/2021/Conference — CoRL2021 Poster_

### Official Review · Reviewer_vtKE · 2021-07-23

**Originality:** Very Good
**Technical Quality:** Excellent
**Clarity Of Presentation:** Very Good
**Impact:** 4

**Recommendation:**

Strong Accept: I recommend accepting the paper and will argue for my recommendation even if other reviewers hold a different opinion.

**Summary:**

This paper proposes a deep learning architecture for off-road terrain traversability. The BEVNet architecture is based on Birds Eye View Network that contains 3 stages; 3D sparse convolutions, a recurrent network and an inpainting network to estimate current and forecast future traversability. The 3D sparse convolutions process voxelized point clouds, the recurrent network accumulates and fuses data over time and the inpatining network completes the traversability map.
Two main datasets are used to evaluate the proposed network, SemanticKITTI and RELLIS-3D. Quantitative, qualitative and ablation study results are presented and benchmarked with state of the art approaches.

**Issues:**

Please define "r" of the feature vector (line 138).
The noise added to odometry is realistic, but it would be good to see how the network behaves with more noise.
In the results I am missing the comparison of BEVNet-S with the vanilla Cylinder-3D. Results could be added to Table 1.
Are the results of Table 2 obtained with the network "full"?
Is there any difference with the end-to-end trained network vs the two-stage training?
In the videos, I noticed there is a flickering issue near the borders in the traversability output of the network. It would good to explain this issue in the RELLIS-3D dataset, where is quite evident.
Finally, it would be relevant for the community if the authors make the network open-source.

**Reviewer Expertise:**

Very good: Comprehensive knowledge of the area

**Strengths And Weaknesses:**

Strengths:
The paper is well written and well structured. The related work seems complete. The methodology is quite clear as well and the results are very well presented.
The authors tackle the problem of semantic mapping and traversability in off-road scenarios, which differs of most of the related research that have focused on on-road developments.
Exploiting geometric and temporal constrains is the key feature of the proposed BEVNet.
The thoroughness of the results is one of the highlights of this paper. The results show very clearly the performance of the network with respect to SOTA and the proposed stages.
The supplementary material helps to a great extent to make results reproducible and to demonstrate the performance.

Weakness
The main weakness in my view is that the network purely rely in data-driven models. Despite the relatively good performance, the authors had to train a separate model per dataset, thus the models have a hard time to generalized. In many circumstances such as data fusion, well established "hand-engineered" models will produce more accurate and robust results in none observed datasets.




**Summary Of Recommendation:**

The authors tackle the relevant problem of off-road navigation, which has been largely overlooked by the community. The proposed network is well justified and explained. The methodology is easy to follow. The analysis of the results show the proposed network performs well in the intended off-road scenarios. The thoroughness of the validation benchmarks very well the network with respect to its variations and the SOTA.

---

> ### Author Response · Authors · 2021-08-27
> **Response to Reviewer vtKE Part 1**
>
> We thank the reviewer for taking the time to write a detailed and insightful review of our paper! Below are our responses to the concerns:
>
> > Weakness The main weakness in my view is that the network purely rely in data-driven models. Despite the relatively good performance, the authors had to train a separate model per dataset, thus the models have a hard time to generalized. In many circumstances such as data fusion, well established "hand-engineered" models will produce more accurate and robust results in none observed datasets.
>
> We agree that the model is very much data-dependent in learning all the geometric/semantic features for the task of traversability map generation, and given certain scenarios can (and sometimes expected) be less data efficient compared to hand-engineered methods. For instance, as we compare our model against baseline JS3C-Net (See our response to Reviewer u6cx Part 1) for simple top-down projection, we observe it is more data-efficient to code the projection method instead of learning it. In more challenging settings such as complex projection, we show our learning-based approach in the 4 category ontology significantly outperforms JS3CNet. Additionally, we want to highlight that our model can use history to better predict the future, whereas traditional SLAM is not able to do so. This is useful in practice as LiDAR scans can get very sparse as distance increases.
>
> We train one model for each dataset mostly for evaluation. In practice, it is possible to combine multiple datasets so that the model can be deployed on a variety of terrains. We trained a model with the two datasets combined and got similar mIoUs (KITTI: 0.531, RELLIS: 0.624) compared to the models trained on each dataset separately (KITTI: 0.535, RELLIS: 0.644).
>
> Additionally, using the model trained on combined datasets we conducted additional experiments where we ran BEVNet-R on data collected with our Warthog platform in various conditions. While the model was not trained in these environments, it shows good generalization capabilities. Please check out the videos on the project website (https://sites.google.com/view/terrain-traversability/home#h.esrg71olrgxo)
>
> > Please define "r" of the feature vector (line 138).
>
> r is the remission value of the LiDAR (see line 139).
>
> > The noise added to odometry is realistic, but it would be good to see how the network behaves with more noise.
>
> As reviewer u6cx and other reviewers have requested, we provide a more detailed ablation study on odometry noise to represent the out-of-control odometry noise condition possible in real robot experiments, included in sec.5 of revised supplementary. We also provide the results here:
>
> |                 | 0\%                       | 50\%    | 100\%   | 200\%   | 500\%   |
> | --------------- | -------                   | ------- | ------- | ------- | ------- |
> | SemanticKITTI   | 0.480                     | 0.474   | 0.468   | 0.458   | 0.431   |
> | RELLIS-3D       | 0.618                     | 0.616   | 0.614   | 0.611   | 0.600   |
>
> While BEVNet-R relatively maintains its performance with increased noise, it tends to retain less history than BEVNet-R trained on clean odometry. This is not surprising because under noisy odometry the history is less reliable. One future direction we would like to pursue is to improve the amount of information retained under noisy odometry. Visualizations of this ablation study are also available on our project website(https://sites.google.com/view/terrain-traversability/home#h.4osy9hqyfe79).

---

> ### Author Response · Authors · 2021-08-27
> **Response to Reviewer vtKE Part 2**
>
> > In the results I am missing the comparison of BEVNet-S with the vanilla Cylinder-3D. Results could be added to Table 1. Are the results of Table 2 obtained with the network "full"? Is there any difference with the end-to-end trained network vs the two-stage training?
>
> Since vanilla Cylinder-3D does not perform any inpainting or temporal aggregation, its mIoU is very poor (mIoU on SemanticKITTI is only 0.11).
>
> The results of Table 2 were obtained when evaluated on the full groundtruth map with clean odometry.
>
> We expect the difference between end-to-end training and two-stage training to be not large. The two-stage training is mainly to save GPU memory. If we train the whole network end-to-end, there will not be enough memory to train with a sufficiently long sequence. Currently, with a RTX-3090 we can train with a sequence of 5 with two-stage training. We expect the performance of our network to improve if trained with a longer sequence.
>
> > In the videos, I noticed there is a flickering issue near the borders in the traversability output of the network. It would be good to explain this issue in the RELLIS-3D dataset, where is quite evident.
>
> The “flickering” is likely due to the LiDAR points being extremely sparse at the border. These seemingly random LiDAR hits at the border would alter how the network inpaints this region. In the RELLIS dataset, the LiDAR was mounted at a lower height (1.2m above the ground) whereas in SemanticKITTI the LiDAR was 1.7m above the ground. Hence, in RELLIS the LiDAR points are more concentrated near the robot, which makes the flickering more noticeable. One potential solution is to add a feedback loop from the output of the network to the recurrent layer so that the network can learn to use the current inpainted map to guide the prediction at the next time step. This would be an interesting future direction to improve BEVNet.
>
> > Finally, it would be relevant for the community if the authors make the network open-source.
>
> We thank the reviewer for their interest. We will open-source our code upon acceptance of our paper.

---

### Official Review · Reviewer_2yv4 · 2021-07-24

**Originality:** Good
**Technical Quality:** Very Good
**Clarity Of Presentation:** Very Good
**Impact:** 4

**Recommendation:**

Weak Accept: I recommend accepting the paper, but will not argue for my recommendation if the majority of other reviewers have a different opinion.

**Summary:**

The paper considers traversability estimation for autonomous off-road driving, and develops Bird-eye View Network (BEVNet) which predicts traversability map from sparse Lidar inputs with semantic information (semantic objects are mapped to labels: free, low-cost, med-cost, lethal). The architecture consists of 3d sparse cnn for point cloud processing, ConvGRU for temporal aggregation of scans, and 2d encoder/decoder to produce traversability birds-eye-view map. As parts of the framework, unseen areas of the map can also be predicted. The traversability map can be converted into a cost map for planning purposes.

The proposed algorithmic aggregates scans, points are semantically mapped to labels, ground plane is estimated, and used as reference for the final projection, before which points of a certain theshold above the ground plane are filtered. For the remaining points, the one with lowest traversability is used as the final 2d projected point label. The vehicle odometry is used to transform and aggregate scans over time.
The network is also trained using noisy perturbed inputs, which might help explain why it is later more robust to noise from odometry (and hence the map aggregation).

The authors include comparisons to Cylinder3d network on two standard datasets -- Kitti and Rellis, demonstrating improved performance.

Other comments:
Would be useful to explain what Cylinder3d does -- it's not clear from the name that it is an appropriate comparison network.
How is the ground truth constructed in the Figures? It seems noiser (and more incomplete) than BEVNet ?

**Issues:**

Would be useful to limit the scope when referring to "off-road" as noted above.
Please clarify and address the comments under Weaknesses above.

**Reviewer Expertise:**

Very good: Comprehensive knowledge of the area

**Strengths And Weaknesses:**

Strengths:
+ general method for semantics+lidar+odometry-based traversability estimation
+ the qualitative results demonstrate that the method is promising
+ the authors have performed comparisons using standard data-sets

Weaknesses:
- the paper claims to address off-road traversability but by explicitly relying on semantics it is extremely difficult to address offroad navigation where aspects such as slope, surface roughness, tire-surface interaction, etc... are not easy to label semantically; the authors might want to limit the scope and be clear that it only applies to a class of environments that can be semantically labeled
- the dependence on a single ground plane might limit the applications to planar environments
- is Cylinder3d the only/best comparison framework?

**Summary Of Recommendation:**

The paper has interesting ideas about combining geometry and semantics for traversability.

---

> ### Author Response · Authors · 2021-08-27
> **Response to Reviewer 2yv4 Part 1**
>
> We thank the reviewer for taking the time to write a detailed and insightful review of our paper! Below are our responses to the concerns:
>
> > How is the ground truth constructed in the Figures? It seems noiser (and more incomplete) than BEVNet ?
>
> The groundtruth is constructed by aggregating the labeled LiDAR scans and projecting them to 2D using the rule described in Sec.3.2. The “noise” may come from the labeled scans themselves as the labels are not perfect (but mostly correct) or the ground plane being unstable (due to sparse LiDAR points). The “incompleteness” is due to the driving pattern of the vehicle and occlusion. BEVNet may look more “complete” because it is free to give predictions to the unknown area. However, when doing the evaluation, we only consider the region where the groundtruth BEV maps have a valid label.
>
> > the paper claims to address off-road traversability but by explicitly relying on semantics it is extremely difficult to address offroad navigation where aspects such as slope, surface roughness, tire-surface interaction, etc... are not easy to label semantically; the authors might want to limit the scope and be clear that it only applies to a class of environments that can be semantically labeled
>
> We agree with the reviewer, and Reviewer 2 (dzpe) shares the same concern. Please see our response (copied below):
>
> We chose 4 levels to represent traversability because 1) it is easier to evaluate and compare with existing methods, and 2) the current publicly available datasets do not contain complex terrains like steep hills. However, the proposed model can be used to learn geometric features in more complex terrains as well. Note that in the labeling process we have access to the complete scene, which is created by aggregating LiDAR scans, and hence the steepness of the terrain at each point can be estimated accurately. Using this information one can define fine-grained categories like flat grass, steep grass, etc., and use these labels to train the BEV net to predict these classes. If necessary, the costmap can contain continuous cost values instead of discrete labels, and BEVNet can be used to directly regress that.
>
> To illustrate this process, we run a simple experiment to utilize the geometric features in the RELLIS-3D dataset. Since the terrain is mostly flat in this dataset, we focus on another useful geometric feature: the height of the vegetation. As the reviewer mentioned, the definition of traversability depends on the robot; vegetation with a certain height might be traversable for a large vehicle such as Warthog while it is not traversable for a smaller robot. With this in mind, we measure the height of bushes during the labeling process. Parts that are taller than a threshold H are marked as lethal obstacles while shorter parts are labeled as medium cost. The height is measured from the closest ground level to the bush. Recall that these measurements are done on the full point cloud and are relatively accurate but the network needs to learn to predict the correct labels from a single LiDAR scan, which only provides a sparse partial observation.
>
> We train 2 different networks with different thresholding: one where the medium cost class is kept whatever its height is, and one where the bush class taller than 0.5m above the ground estimate is determined non-traversable and hence remapped to lethal. We visualize the network predictions on the test set, example predictions provided in Sec.6 of the revised supplementary and on the project website (https://sites.google.com/view/terrain-traversability/home#h.5c7sml45lt56). The results show that as we decrease the height the model becomes more conservative in marking bushes as medium-cost traversable.
>
> > the dependence on a single ground plane might limit the applications to planar environments
>
> We do not assume a single ground plane when generating the ground truth BEV maps. The local ground level is estimated by averaging the traversable points with the lowest z values at each x, y location (see Sec. 3.2, Points Binning and Ground Plane Estimation). Hence the ground level can vary between different locations. We have revised L128-L132 in the section for clarity.

---

> ### Author Response · Authors · 2021-08-27
> **Response to Reviewer 2yv4 Part 2**
>
> > is Cylinder3d the only/best comparison framework?
>
> Cylinder3D is a neural network for LiDAR segmentation, and it was specifically chosen as the segmentation method to represent the semantic segmentation component of the pipeline. It is also the most recent SOTA (CVPR21) with code available. The segmented point cloud from Cylinder3D is then postprocessed based on our projection rule and aggregated in 2D to generate the BEV costmap.
>
> Per the request of Reviewer 2 (dzpe) we also include an additional baseline that does 3D semantic aggregation. Please see our response to Reviewer 2 [W5] and also Sec. 7 of the supplementary material for the results and discussions.
>
> We also make additional comparisons against other baselines on the full categories per the request of Reviewer 1 (u6cx). Note that we have to compare with these baselines by tailoring our model to their settings (smaller maps, full categories, topdown projection), and the results do not fully translate to our task of terrain traversability estimation.

---

### Official Review · Reviewer_dzpe · 2021-07-26

**Originality:** Good
**Technical Quality:** Very Good
**Clarity Of Presentation:** Good
**Impact:** 3

**Recommendation:**

Weak Accept: I recommend accepting the paper, but will not argue for my recommendation if the majority of other reviewers have a different opinion.

**Summary:**

The authors propose a method for LiDAR semantic information aggregation and completion. The target application of their method is for off-road traversability prediction.

**Issues:**

Please address/comment on/rebut the weakness listed above or reframe the work as squarely on semantic information fusion, or demonstrate the applicability of the approach by deploying it on a real robot (probably not possible in the rebuttal time frame).

**Reviewer Expertise:**

Good: General knowledge of the area

**Strengths And Weaknesses:**

Strengths:

The authors have constructed a fairly complex system for aggregating semantic information from LiDAR data. The odometry integration is differentiable so that gradients can be backpropagated through time to perform the aggregation. The sparsity of the LiDAR data is overcome partially through this aggregation but also through an in-painting network. The results show that they are able to produce semantic maps over their 4-class ontology that are stable could be useful for off-road robot navigation.

Weaknesses:

[W1] I feel the largest weakness of the work is that the approach is designed for predicting terrain traversability, but the work never deploys the model to evaluate navigation on a mobile robot. Some of the choices that are made about what classes should be deemed "traversible" and which classes not will be traversible depends heavily on the specific robot. I'm not sure it will always be clear that some classes should be traversible and others not in fact. It would seem that a requirement for demonstration of the value of their traversability prediction is that it is in fact traversible for some specific robot. Without this, the method is just aggregating semantic information and fusing it with odometry.

[W2] I'm not clear on whether geometry is being used to determine traversability at all. On line "For the remaining points, we take the point with the lowest traversability at each x, y location as the final traversability label" it seems not. It would seem that for traversability we need a mix of geometry and semantic information. For example, if grass is a steep hill then it may not be traversible but if it is flat it will be.

[W3] I'm somewhat confused about how the aggregation is accounting for odometry error both in the dataset generation and at inference time. On L 211 you claim that "BEVNet-R uses learned recurrency to “fix” the errors in odometry" but I don't understand what this means or even how this is possible. Please clarify/elaborate.

[W4] I find the 3rd claim of contribution overstated. The claim is " 3) filtering out irrelevant obstacles like overhanging tree branches that do not affect traversability" which is achieved in the methodology by "we filter out overhanging obstacles by remov- ing points that are above the ground plane by a certain threshold because they will not collide with the robot." This doesn't seem to me to warrant a claim of novelty.

[W5] I would have like to have seen a comparison against a more traditional SLAM-based fusion approach.

**Summary Of Recommendation:**

It is difficult to assess the true usefulness of this approach for real robotics navigation since the model is never shown to work in this setting. The ontology that is used to define traversability is hand-constructed and it is not clear (at least to me) to what degree this output actually maps to traversability of a real robot in a practical setting. It is possible that the work could be re-framed as an approach for semantic fusion and completion for LiDAR data, but as it stands it is hard to evaluate on the use-case that it sets out to achieve, which is traversability.

-----------------------

Post-rebuttal comments.

The authors have done an exceptional job trying to address the comments that I raised in the first round. I particularly appreciate the new comparisons with a SLAM-based baseline. I have correspondingly updated my recommendation to "Weak accept".

A few further suggestions for the camera-ready version if the paper is accepted. I would suggest to move some of the responses to my (and other reviewers') comments out of the supplementary and into the main manuscript. For example, in my opinion, the comparison with the traditional SLAM approach is much more valuable/insightful than the discussion about early/late aggregation in the ConvGRU. I would also suggest that they update the description of the third claim of novelty to include more of the details in the response here since as stated in the manuscript currently it seems kind of trivial, but only now do I understand that it is really not. Additionally, if the authors' have performed experiments on the Warthog (as evidenced by the project page), it would strengthen the paper to include at least some qualitative evidence of this to strengthen this as a truly applicable and battle-tested method for robotics.

---

> ### Author Response · Authors · 2021-08-25
> **Response to Reviewer dzpe  W1, W2**
>
> > [W1] I feel the largest weakness of the work is that the approach is designed for predicting terrain traversability, but the work never deploys the model to evaluate navigation on a mobile robot.
>
> We thank the reviewer for pointing this out. For offroad traversability analysis we base our experiments on the RELLIS-3D dataset, which is collected using a Warthog robot that is "a large all-terrain unmanned ground vehicle capable of brief periods of locomotion in water. It can handle tough environments with its rugged build, low ground pressure, and traction tires, which allow effortless mobility through soft soils, vegetation, thick muds, and steep grades" [R1]. Traversability classification is typically done at binary level (traversable/ non-traversable), while several works use fine-grained categories to define traversability costs. Schilling et al.[R2] proposes 3 categories: safe, risky, and obstacle. Maturana et. al.[R3] uses a more varied set of semantic classes: void, vegetation, tree, trail, etc. Given the context application of navigation on the Warthog, the parsimonious categorization can be free (e.g. road), low-cost (e.g. low vegetation), medium cost (e.g. bush, traversable but potentially risky), obstacle. As we show in our response to [W2], this categorization can be modified or expanded based on the robot requirements and using geometric features, if necessary.
>
> Through several experiments on the Warthog, we realized that 4 categories are sufficient for successful navigation in many on-road and off-road scenarios. For better illustration and showing the testing conditions of the Warthog, we have included several videos on the project website showing the testing condition of the proposed system on the Warthog (https://sites.google.com/view/terrain-traversability/home#h.esrg71olrgxo").
>
> In one of the videos (weeds), we highlight the fact that the robot can traverse over dense vegetation that is categorized as the medium-cost group. While it is still traversable, it is naturally less preferable and is a potentially risky terrain to drive over. This additional level of traversability allows the robot to decide if it prefers to traverse through the medium-cost area to get to the target quickly or find an alternative route at a lower risk.
>
> [R1] Clearpath Warthog unmanned ground vehicle description page, https://clearpathrobotics.com/warthog-unmanned-ground-vehicle-robot/
>
> [R2] Schilling et al. "Geometric and visual terrain classification for autonomous mobile navigation." IROS 2017.
>
> [R3] Maturana et al. “Real-Time Semantic Mapping for Autonomous Off-Road Navigation.” FSR 2017.
>
>
> > [W2] I'm not clear on whether geometry is being used to determine traversability at all.
>
> We chose 4 levels to represent traversability because 1) it is easier to evaluate and compare with existing methods, and 2) the current publicly available datasets do not contain complex terrains like steep hills. However, the proposed model can be used to learn geometric features in more complex terrains as well. Note that in the labeling process we have access to the complete scene, which is created by aggregating LiDAR scans, and hence the steepness of the terrain at each point can be estimated accurately. Using this information one can define fine-grained categories like flat grass, steep grass, etc. and use these labels to train the BEV net to predict these classes given a single LiDAR scan.
>
> To illustrate this process, we run a simple experiment to utilize the geometric features in the RELLIS-3D dataset. Since the terrain is mostly flat in this dataset, we focus on another useful geometric feature: the height of the vegetation. As the reviewer mentioned, the definition of traversability depends on the robot; vegetation with a certain height might be traversable for a large vehicle such as Warthog while it is not traversable for a smaller robot. With this in mind, we measure the height of bushes during the labeling process. Parts that are taller than a threshold H are marked as lethal obstacles while shorter parts are labeled as medium cost. The height is measured from the closest ground level to the bush. Recall that these measurements are done on the full pointcloud and are relatively accurate but the network needs to learn to predict the correct labels from a single LiDAR scan, which only provides a sparse partial observation.
>
> We train 2 different networks with different thresholding: one where the medium cost class is kept whatever its height is, and one where the bush class taller than 0.5m above the ground estimate is determined non-traversable and hence remapped to lethal. We visualize the network predictions on the test set, example predictions provided in Sec.6 of the revised supplementary and on the project website (https://sites.google.com/view/terrain-traversability/home#h.5c7sml45lt56). The results show that as we decrease the height the model becomes more conservative in marking bushes as medium cost traversable.

---

> ### Author Response · Authors · 2021-08-25
> **Response to Reviewer dzpe W3, W4**
>
> > [W3] I'm somewhat confused about how the aggregation is accounting for odometry error both in the dataset generation and at inference time.
>
> In both KITTI and RELLIS, the scans are accurately aligned via the poses computed by a SLAM pipeline. We use these accurate poses to generate the ground truth traversability maps, so the ground truth has minimal odometry error. During training, we optionally inject noise to the odometry (part of the input), but still supervise the model with error-free ground truth traversability maps. To minimize the loss, the network has to learn to be robust to noisy odometry. For small odometry errors, the network can learn to correct that since both the ConvGRU and the Inpainting Network have convolutional kernels which look at the neighborhood of each pixel to fix small misalignment. For large odometry errors, the network can decide to forget part of the history. Our quantitative results in the paper show that with noisy odometry, BEVNet-R outperforms BEVNet-S, BEVNet-S-TA and Cylinder3D-TA. This shows that BEVNet-R is capable of adaptively retaining the history. We additionally provide ablation studies on the different levels of odometry noise during testing for BEVNet-R, included in Sec.4 of the revised supplementary (also in the table below) with visualization videos uploaded to the project website (https://sites.google.com/view/terrain-traversability/home#h.4osy9hqyfe79). We observe that BEVNet-R is robust to various levels of noise, and the alignment correction and forgetting history can be observed from the visualizations.
>
> |                 | 0\%                       | 50\%    | 100\%   | 200\%   | 500\%   |
> | --------------- | -------                   | ------- | ------- | ------- | ------- |
> | SemanticKITTI   | 0.480                     | 0.474   | 0.468   | 0.458   | 0.431   |
> | RELLIS-3D       | 0.618                     | 0.616   | 0.614   | 0.611   | 0.600   |
>
> One thing we would like to point out is that BEVNet-R trained on noisy odometry tends to retain less history than BEVNet-R trained on clean odometry. This is not surprising because under noisy odometry the history is less reliable. One future direction we would like to pursue is to improve the amount of information retained under noisy odometry.
>
> > [W4] I find the 3rd claim of contribution overstated (filtering out irrelevant obstacles)
>
> We disagree with the reviewer about the novelty of this approach. First note that the filtering mechanism that removes the irrelevant obstacles like overhanging branches does not work well if it is applied on a single parse LiDAR scan as the ground level can not be accurately estimated. We do the filtering in the ground truth label generation process where the complete scene is available and therefore the ground level estimation can be done accurately. By using such ground truth labels without the actual filtering mechanism, BEVNet learns to perform the necessary inpainting, pattern recognition, and removal of overhanging structures from sparse LiDAR scans to predict accurate BEV maps that match the ground truth.
>
>
> the network has to learn to filter such obstacles in the 3D space to predict accurate BEV maps that match the ground truth. For this, we expect the network to do a lot of inpainting and pattern recognition to successfully identify and remove the overhanging structures from a sparse LiDAR scan. We believe that the novelty is in the learning process that allows BEVNet to filter such obstacles without the actual mechanism or information provided.
>
>
> > [W5] I would have like to have seen a comparison against a more traditional SLAM-based fusion approach.
>
> We are in the process of implementing a traditional SLAM-fusion baseline and will add the result once it is completed.

---

> ### Author Response · Authors · 2021-08-26
> **Response to Reviewer dzpe W5**
>
> We thank the reviewer for taking the time to write a detailed and insightful review of our paper. Applying your suggestions indeed improves the quality of our paper. Here is our response to the last concern [W5]:
>
> > [W5] I would have like to have seen a comparison against a more traditional SLAM-based fusion approach.
>
> As the reviewer requested, we provide comparison results of our model against a 3D SLAM-fusion baseline (called Cylinder3D-TA-3D). Here is how this baseline works:
>
> 1. We first perform semantic segmentation on the LiDAR scans with Cylinder3D to generate segmented point clouds.
>
> 2. We aggregate the segmented point clouds in 3D using a voxel grid similar to how Octomap [R1] works using calibrated poses from SemanticKITTI and RELLIS-3D.
>
> 3. When aggregating the point clouds, we optionally apply ray tracing to remove the trace of moving objects as much as possible.
>
> 4. For the points falling into a voxel, we normalize the counts of their predicted labels into a categorical distribution. This is how Bayesian statistics estimates the parameter of a categorical distribution via a uniform Dirichlet prior. This produces a labeled voxel grid.
>
> 5. We project this labeled voxel grid down to 2D using the same rule as how we generate the groundtruth BEV costmaps.
>
> We refer the reviewer to Sec.7 of the supplementary for complete results and detailed analysis. We summarize the main points here:
>
> 1. Compared to Cylinder3D+TA baseline that aggregates in the 2D BEV space, 3D aggregation with raytracing exhibits considerable improvement in the seen area: +8 points in RELLIS (+5.9 points in SemanticKITTI), respectively. However, while 2D aggregation time in TA is negligible, 3D aggregation is the computational bottleneck in the pipeline as it takes ~1 sec to aggregate a single scan into the octomap and perform ray tracing.
>
> 2. When evaluating on the whole map, BEVNet outperforms Cylinder3D-TA-3D by more than 20 points in RELLIS (5 points in SemanticKITTI) because Cylinder3D-TA-3D does not predict the future.
>
> 3. When limiting the comparison to the seen area, Cylinder3D-TA-3D outperforms BEVNet-R up to 3 points in RELLIS (6 points in SemanticKITTI).
>
> In conclusion, there is a trade-off between prediction area and accuracy. While combining state-of-the-art semantic segmentation with SLAM-based fusion produces better results on the seen area, it falls short of predicting a more complete map, is prone to odometry noise, and is computationally expensive. Our network is optimized to perform multiple tasks simultaneously (semantic segmentation, inpainting, complex projection, time aggregation) efficiently in a single forward pass.
>
> [R1] Hornung, Armin, et al. "OctoMap: An efficient probabilistic 3D mapping framework based on octrees." Autonomous robots 2013

---

### Official Review · Reviewer_u6cx · 2021-07-26

**Originality:** Good
**Technical Quality:** Fair
**Clarity Of Presentation:** Good
**Impact:** 3

**Recommendation:**

Weak Accept: I recommend accepting the paper, but will not argue for my recommendation if the majority of other reviewers have a different opinion.

**Summary:**

The submission proposes a method for predicting dense traversability maps from LIDAR/pointcloud inputs. The method applies a deep model, various tricks for handling sparse data and is trained via supervised learning. The evaluation is performed on variants of 2 benchmarks (SemanticKITTI and RELLIS-3D) and includes the comparison against one external baseline and a set of variations of the proposed method.

**Issues:**

- extremely limited external baselines.
- not evaluated on complete versions of the applied benchmarks (if there is a good argument for why this is not feasible, I can drop this point).
- further issues in evaluation as described in the main review.


**Reviewer Expertise:**

Good: General knowledge of the area

**Strengths And Weaknesses:**

The paper is mostly clearly written and structured. The proposed method outperforms the introduced baseline and additional ablations are useful to understand various parts of the method. In particular, the questions addressed in section 5.2.1 help the reader to better disentangle the individual contributions of various modelling choices.

There are a couple of open questions regarding this paper. A key point in the method seems to be the reduction to 4 traversability categories instead of the original set of categories in existing datasets. It is not clear to me why the presented method could not be applied to the full set of categories and, related, be compared against the full set of methods which have been applied to these datasets. This would not need to replace the main evaluation but would be an important extension for comparing against the existing state of the art in semantic segmentation in these domains. It is even more important as the evaluation only uses a variant of a single external baseline. It would be easy to make space in the paper for an additional comparison by moving the implementation details to the appendix (section 4).

For the method itself there remain some unclear aspects of which I’ve added most under minor issues below. One specific question is how the unknown class is treated. The paper describes its use for evaluation, but couldn’t we also use it for training the network? Since the unknown class depends on what happens in the future, my intuition would be that it is impossible (or at least very hard) to predict as it will depend on driving behaviour. In addition, if predicting the unknown class is never informative for evaluation, it might make sense to not train the model in the first place on cells which are labelled as unknown.

Another question regarding the evaluation is the test with noisy odometry, a key area where the proposed method outperforms the external baseline. Since you control the noise during training of your model and during evaluation, it would be important to ablate this aspect. In particular if you have defined the noise to be the same across both. Real odometry noise would be out of your control and it would be good to see how good predictions are if your training noise exceeds or is exceeded by the evaluation noise.

Minor issues:
- L35: ‘efficient backbone’ could be described in a more informative way; how does inpainting work here. Short description would suffice.
- L38: It would help to start with stating that this relies on ground-truth labels in the datasets. There has been a good share of related work which simply uses trajectories (ie odometry) and inverse reinforcement learning to determine traversability maps [1,2]
- Section 4: why do the different datasets require very different numbers of frames and stride? Were these collected at different frequencies? If so it would be good to add that information to justify changes in hyperparameters.
- L176: referring to the subsection would be helpful (instead of just Sec. 5)
- L179: Out of curiosity: Would it be possible to train a single model on both datasets?
- L205: ‘C3D-TA’ Please use the same abbreviations in table and text.

[1] Kris M. Kitani, Brian D. Ziebart, James Andrew Bagnell, and Martial Hebert. Activity  Forecasting. Computer Vision ECCV 2012
[2] Wulfmeier, Markus, Dominic Zeng Wang, and Ingmar Posner. "Watch this: Scalable cost-function learning for path planning in urban environments." 2016 IEEE/RSJ International Conference on Intelligent Robots and Systems (IROS). IEEE, 2016.


**Summary Of Recommendation:**

The submission presents a mostly well-written and intuitively structured paper. The method is new but close to existing approaches and the application is relevant. The overall impact of the paper is likely within limits but could be considerably increased by extending the comparison wrt more baselines (currently only one) and the use of further semantic classes by using the full version of both applied benchmarks.

(rebuttal response: For reasons described during the rebuttal discussion, I will increase the score.)

---

> ### Author Response · Authors · 2021-08-23
> **Response to Reviewer u6cx Part 1**
>
> We thank the reviewer for taking the time to write a detailed and insightful review of our paper. Applying your  suggestions indeed improves the quality of our paper. We will update our paper to incorporate all your comments. In light of that, we would greatly appreciate it if you would consider increasing your score to “accept”.
>
> > It is not clear to me why the presented method could not be applied to the full set of categories and, related, be compared against the full set of methods which have been applied to these datasets.
>
> ### Why we prefer 4-level traversability classification
>
> When using the full set of categories, BEV projection can be ambiguous when more than one category is present in a single location (e.g. person driving a bike) of the BEV map. To solve this ambiguity most methods (Han et al. [7]) do a top-down projection in which the point with the highest z value is used to label the map. Although this method is simple and does not require learning, it is not suitable for traversability analysis in off-road scenarios because the highest point does not necessarily specify the traversability. In our 4 class traversability ontology, this ambiguity can be easily addressed by giving priority to the class with the highest traversability cost.
>
> Categorization by traversability cost rather than semantic classes is common in terrain traversability analysis literature [R1]. Most of the works utilize a binary classification ontology (traversable/ non-traversable). Schilling et al. [R2] propose 3 categories: safe, risky, and obstacle. Given the context application of navigation of Warthog the parsimonious categorization is seen to be free (e.g. road), low-cost (e.g. low vegetation), medium cost (e.g. bush, traversable but potentially risky), obstacle.
>
> [R1] Guastella et al. "Learning-Based Methods of Perception and Navigation for Ground Vehicles in Unstructured Environments: A Review." Sensors 2021.
>
> [R2] Schilling et al. "Geometric and visual terrain classification for autonomous mobile navigation." IROS 2017.
>
> ### Comparison in the full set of categories
>
> Following is the response to the request comparing with the SOTA baselines on the original set of categories. We are providing the results (Table 8 in the supplementary) for the SOTA semantic segmentation Cylinder-3D (CVPR21) in full categories along with the SOTA inpainting methods JS3CNet (AAAI21), and Han et al. [7] BEV map inpainting method (CoRL20). The implementation details can be found in Sec.4 of the supplementary. An off-road autonomous navigation system should additionally detect overhanging structures, and perform time aggregation. To highlight this, we also show the tasks each model does in the table. Note that these baselines produce much smaller maps than ours, so we need to tailor our model to their settings.
>
> |Method(\#classes)|Segmentation|Inpainting|Projection|Temporal Aggregation|seen|unseen|
> |--------------------|--------------|------------|------------|----------------------|------------|------------|
> |Cylinder3D-TA(19)|Yes|No|TopDown|Yes|**0.316**|0.181|
> |Han et al.(19)|No|Yes|TopDown|No|-|0.131|
> |JS3C-Net(19)|Yes|Yes|TopDown|No|-|**0.258**|
> |BEVNet-S(19)|Yes|Yes|TopDown|No|0.313|0.253|
> |JS3C-Net(4)|Yes|Yes|Complex|No|-|0.549|
> |BEVNet-R(4)|Yes|Yes|Complex|Yes|-|**0.608**|
>
> **Results**: When testing on full categories and using top-down projection, JS3CNet is slightly better than our method, specifically for classes within the lethal obstacles group. Note that in our approach, the projection is learned from the data and not hand-engineered as in top-down projection. We hypothesize for simple top-down projection, it is more data-efficient to code the projection method instead of learning it from the data. However, in complex projection, our learning-based approach in the 4 category ontology significantly outperforms JS3CNet.
>
> Speedwise, JS3C-Net is significantly slower than ours since it predicts a dense voxel grid. On a 51.2m x 51.2m map, JS3C-Net runs less than 2 fps whereas BEVNet-S runs at 12 fps on a 1080 Ti. It will be hard to scale JS3C-Net to larger maps with recurrency (note that in our main results we use 102.4m x 102.4m maps). In comparison, BEVNet maintains a latent 2D feature map for cost reasoning, which is more memory-efficient.
>
> Cylinder 3D's predictions are limited to the seen area of the map. It is surprising that our method performs as well as the Cylinder 3D on the seen area while being able to inpaint the entire map. Finally, we emphasize that our network is designed to perform multiple tasks simultaneously (semantic segmentation, inpainting, complex projection, time aggregation). It is not surprising that its performance does not exceed Cylinder-3D or JS3CNet which perform a subset of these tasks using a network with a similar capacity.

---

> > ### Comment · Reviewer_u6cx · 2021-09-02
> > **Rebuttal discussion**
> >
> > Thank you for addressing the feedback. I highly appreciate the additional experiments on noise, combined datasets and semantic classes as well as additional information.  In particular, the inclusion of the original semantic classes which considerably extends the set of related work and baselines to correctly integrate this work into the field is highly appreciated. I'm uncertain about the described shortcomings and would render the emphasised challenges as more of minor impact. The increased score can be found in the main review.

---

> ### Author Response · Authors · 2021-08-23
> **Response to Reviewer u6cx Part 2**
>
> > One specific question is how the unknown class is treated.
>
> We agree the definition of unknown class mostly depends on the driving behavior and it is impossible to be learned if driving trajectories do not have a predominant pattern. In practice, we observe a lane following behavior by the driver, either on the road or on the trail, in KITTI/RELLIS datasets and the points that are close to the driver’s trajectory are more likely to be seen in the future. Thus, the network output can be visualized better by marking the points that are far from the future trajectory as unknown. Han et al. [7] also utilize this for better visualization and further applies convex hull on the current LiDAR scan to mark regions outside the fit to mark as unknown, based on the realization that regions without any nearby points are difficult for an inpainting algorithm to accurately estimate as no contextual information is provided.
>
> In our implementation, we represent this by marking unobserved locations from LiDAR data along the driven trajectory, following a similar but simpler methodology than the semantic scene completion benchmark in SemanticKITTI [12]. We remark that the unknown class is only used for visualization.
>
> > Real odometry noise would be out of your control and it would be good to see how good predictions are if your training noise exceeds or is exceeded by the evaluation noise.
>
> As the reviewer requested, we evaluate our model on different levels of odometry noise during testing and include the results in Sec.4 of the revised supplementary. Based on the suggestion we use various scales of noise (0%, 50%, 100%, 200%, 500%) to consider evaluation noise exceeding the training noise and vice versa. The model is trained at 100% noise level. Performance of BEVNet-R degrades gracefully as odometry noise increases compared to the training odometry noise. For qualitative results, we have uploaded videos to the project website showing the predictions at different levels of odometry noise.
>
> |                 | 0\%                       | 50\%    | 100\%   | 200\%   | 500\%   |
> | --------------- | -------                   | ------- | ------- | ------- | ------- |
> | SemanticKITTI   | 0.480                     | 0.474   | 0.468   | 0.458   | 0.431   |
> | RELLIS-3D       | 0.618                     | 0.616   | 0.614   | 0.611   | 0.600   |
>
> > why do the different datasets require very different numbers of frames and strides?
>
> For the ground-truth labels, we choose the number of frames for accumulation such that it covers the whole area of the map (~100x100 meters) while setting a stride for efficiency. The number of frames we need to cover the map depends on the speed of the vehicle, the height of the LiDAR, and the number of points in each scan so we chose different settings for each dataset.
>
> > Would it be possible to train a single model on both datasets?
>
> We agree it is possible to train a single model on both datasets. When doing so, we also need to make sure that the remission values in both datasets are normalized to be in the same range as different LiDAR sensors are used in each dataset. With the two datasets combined, we get similar mIoUs (KITTI: 0.531, RELLIS: 0.624)) compared to the models trained on each dataset separately (KITTI: 0.535, RELLIS: 0.644).

---

### Meta-Review · Area_Chair_J9vf · 2021-08-13

**Recommendation:** Accept (Poster)
**Confidence:** 5

**Metareview:**

The paper proposes an approach for traversability estimation from LiDAR point clouds. The reviewers find the paper in general to be clearly written and well structured. The reviewers appreciate the ablation study. Most of the major concerns raised by the reviewers have been addressed in the rebuttal. I thank the authors for incorporating all the suggestions and I recommend acceptance.

---

> ### Author Response · Authors · 2021-08-28
> **Response to AC**
>
> We thank the area chair for pointing out the main strengths and issues raised by the reviewers. We are excited that two reviewers rated for acceptance of the paper. We devote the rebuttal to addressing all the reviewers' concerns and requests. Addressing these issues helped us improve the paper.
>
> > Additional baseline/ comparison with SLAM-based fusion
>
> We have added an additional baseline (Cylinder3D-TA-3D) that performs 3D temporal aggregation of semantics using traditional SLAM-fusion techniques (Octomap). We show that while this 3D baseline outperforms our previous baseline and can work very well in the seen area, it is not able to predict the future and is computationally expensive. Additional details can be found in our responses to Reviewer dzpe [W5] and Sec.7 of the supplementary. We have also added results on full categories for SOTA inpainting methods JS3CNet (AAAI21), and Han et al. BEV map inpainting method (CoRL20). Additionally, we added several videos to the project website showing the performance of the proposed algorithm on the Warthog ground vehicle in real-world scenarios.
>
> > Traversability definition
>
> We agree with the reviewers that there are additional geometric factors that can affect traversability. We mainly use semantics to determine traversability for this work because these publicly available datasets (SemanticKITTI and RELLIS-3D) contain mostly flat terrains. We add an additional experiment to show that our labeling process can be simply adapted to utilize geometric information (the height of bushes) that affects the traversability (See our responses to Reviewer dzpe  [W1, W2] and also Sec.6 of the supplementary). This shows BEVNet can use geometrical features along with semantic information to reason about the traversability.
>
> > the evaluations seem to only use a subset of the classes in the benchmark
>
> As the reviewer requested, we provide results on the full category for the SemanticKITTI dataset in Sec.4 of the supplementary. We explain in Part 1 of our response to Reviewer u6cx why we still believe compressing the full semantic categories into 4 traversiblity classes makes sense for our application. To summarize, when projecting to 2D:
> * When using full categories, if 2 obstacles of different classes co-exist at the same location, there is no exact order of preference.
> * When using an ontology directly based on traversability, the costmap can hold various costs for the same class e.g. bushes depending on the height.
>
> Because of this, we think an ontology directly related to terrain cost makes more sense. BEVNet can predict robot-dependent traversability maps given correct supervision (Sec.6 of the supplementary).
>
> > Limitations to planar environments
>
> As we pointed out in response to Reviewer 2yv4 part 1, BEVNet does not assume a flat terrain. We have revised L128-132 for further clarification.

---

> > ### Comment · Area_Chair_J9vf · 2021-08-30
> > **Response to rebuttal**
> >
> > Thanks to the authors for the responses. I still have a concern regarding using the term traversability for defining what the proposed approach is doing. In my opinion, it's more of "terrain classification" than traversability estimation. Simply classifying into a set of predefined categories is insufficient for traversability estimation in the real world. The novelty of the approach is itself limited if you say that the approach is performing traversability estimation since the contribution is more for classification. I would advise the authors to think about rephrasing the title and goals focusing more on classification.
> >
> > How would the approach generalize to environments that are previously unknown in terms of having varying heights of obstacles. The authors have mentioned that the heights of the obstacles can be included in the labeling. However, in unstructured environments such as trails and forests, the amount of variation cannot be predetermined. So a pure classification approach is insufficient to determine the traversability in environments that are different from the training data, in this case, different from the RELLIS-3D and SemanticKITTI.

---

> > > ### Author Response · Authors · 2021-08-31
> > > **Response to AC**
> > >
> > > We thank the Area Chair for their response. We updated the title and manuscript to i) better motivate the paper by explaining that we focus on terrain classification for traversability assessment and cite prior works that treat this problem as terrain classification; and ii) explain in the conclusion the limitations of current approach and how it can be extended. We also made various changes to clarify that we are performing classification on the traversability of terrains. Any edits on the manuscript have been highlighted blue. These changes do not affect the underlying technical content and the experiment section of the paper.
> > >
> > > Specifically, we formulate traversability estimation as a semantic classification problem. We agree that this task formulation can limit the approach regarding traversability estimation beyond classification. The motivation is to unify the semantics (what objects are present?) and geometry (where are the objects located?) of the terrain into a single cost ontology. We use a discrete set of traversability levels for the ease of grouping the semantic classes by their traversability while allowing us to adjust the traversability levels of specific instances based on their geometry. We find this makes it easier to comprehend and to compare with existing methods. Also, note that the publicly available datasets used to train BEVNet do not contain rough terrains, so it is hard to show if taking additional terrain features would improve the overall system. When needed, BEVNet can be modified without major architectural changes to predict continuous costs or fuse the predicted terrain classes with an elevation map to incorporate additional terrain features.
> > >
> > > Regarding generalization, it is generally true that data-driven methods may not generalize to test data that is substantially different from training data. The generalization gap would depend on the quantity and diversity of the training data, which can be closed when more training data is available. Our experiments on a real Warthog robot show that BEVNet generalizes well to several environments that are considerably different from the training set (https://sites.google.com/view/terrain-traversability/home#h.esrg71olrgxo).

---

### Decision · Program_Chairs · 2021-09-13

**Decision:**

Accept (Poster)

**Comment:**

The paper proposes an approach for traversability estimation from LiDAR point clouds. The reviewers find the paper in general to be clearly written and well structured. The reviewers appreciate the ablation study. Most of the major concerns raised by the reviewers have been addressed in the rebuttal. I thank the authors for incorporating all the suggestions and I recommend acceptance.